

# Predicting gene expression using DNA methylation in three human populations

Huan Zhong[1], Soyeon Kim[4], Degui Zhi[2] and Xiangqin Cui[3]

[1] Department of Biology, Hong Kong Baptist University, Hong Kong, China
[2] School of Biomendical Informatics, University of Texas Health Center at Houston, Houston, TX, United States of America
[3] Department of Biostatistics and Bioinformatics, Emory University, Atlanta, GA, United States of America
[4] School of Medicine, University of Pittsburgh, Pittsburgh, PA, United States of America

## ABSTRACT

**Background**. DNA methylation, an important epigenetic mark, is well known for its regulatory role in gene expression, especially the negative correlation in the promoter region. However, its correlation with gene expression across genome at human population level has not been well studied. In particular, it is unclear if genome-wide DNA methylation profile of an individual can predict her/his gene expression profile. Previous studies were mostly limited to association analyses between single CpG site methylation and gene expression. It is not known whether DNA methylation of a gene has enough prediction power to serve as a surrogate for gene expression in existing human study cohorts with DNA samples other than RNA samples.

**Results**. We examined DNA methylation in the gene region for predicting gene expression across individuals in non-cancer tissues of three human population datasets, adipose tissue of the Multiple Tissue Human Expression Resource Projects (MuTHER), peripheral blood mononuclear cell (PBMC) from Asthma and normal control study participates, and lymphoblastoid cell lines (LCL) from healthy individuals. Three prediction models were investigated, single linear regression, multiple linear regression, and least absolute shrinkage and selection operator (LASSO) penalized regression. Our results showed that LASSO regression has superior performance among these methods. However, the prediction power is generally low and varies across datasets. Only 30 and 42 genes were found to have cross-validation $R^2$ greater than 0.3 in the PBMC and Adipose datasets, respectively. A substantially larger number of genes (258) were identified in the LCL dataset, which was generated from a more homogeneous cell line sample source. We also demonstrated that it gives better prediction power not to exclude any CpG probe due to cross hybridization or SNP effect.

**Conclusion**. In our three population analyses DNA methylation of CpG sites at gene region have limited prediction power for gene expression across individuals with linear regression models. The prediction power potentially varies depending on tissue, cell type, and data sources. In our analyses, the combination of LASSO regression and all probes not excluding any probe on the methylation array provides the best prediction for gene expression.

Corresponding author
Xiangqin Cui,
xiangqin.cui@emory.edu

## BACKGROUND

DNA methylation has long been recognized as an important epigenetic modification in regulating gene expression (*Razin & Riggs, 1980*). This process often occurs at CG dinucleotides sites (CpG sites), adding a methyl group to the cytosine residue (*You & Jones, 2012*). In mammalian genomes, more than 70% of CpG sites are methylated (*Jabbari & Bernardi, 2004*). Many CpGs are clustered into CpG islands and more than 30,000 CpG islands have been identified in the human genome, most of which are located in promoter region and are hypo-methylated (*Jeziorska et al., 2017*). The level of DNA methylation at a CpG site is often correlated with that of neighbouring CpG sites and influenced by other genome features, such as genome position and regulatory elements. When combined, these genome features can effectively predict methylation level of CpG sites in the genome (*Zheng et al., 2017*).

The regulatory role of DNA methylation on gene expression has traditionally been studied with a small number of CpG sites in a limited number of genes. The more recent application of microarrays and next generation sequencing enables large-scale analysis of DNA methylation and gene expression across the whole genome (*Krueger et al., 2012*). However, most human genome-wide methylation and expression studies in non-cancer tissues have small sample sizes for comparing controlled groups. Only a limited number of studies profiled both genome-wide DNA methylation and gene expression in larger human populations and examined their relationship across individuals. *Del Rey et al. (2013)* studied the genome-wide DNA methylation and gene expression in 83 low-risk subtypes of Myelodysplastic syndrome (MDS) patients and 36 controls using microarrays. They found negative correlations between methylation and gene expression across individuals in a large proportion of differentially expressed and differentially methylated genes, but they also uncovered substantial positive correlations. In another study of 648 twins, overall negative correlations were found in the adipose tissue, promoter region (−0.018), gene body (−0.013) and 3-prime UTR (−0.007) (*Grundberg et al., 2013*). More recently, *Wagner et al. (2014)* profiled the genome wide DNA methylation and gene expression in forearm skin fibroblast among 62 unrelated individuals. They observed that the association between gene expression and methylation is not always negative in promoter region or positive in gene body (*Yang et al., 2014*).

The complex relationship among DNA methylation, gene expression, and genetic variants in human populations has also attracted substantial research attention. *Bell et al. (2011)* investigated the genetic controls for both methylation QTL (mQTL) and expression QTL (eQTL) using 77 human lymphoblastoid cell lines (LCLs) from the HapMap collection. They identified hundreds of mQTLs and eQTLs and showed that these two types of QTLs overlap significantly. *Gutierrez-Arcelus et al. (2013)* further examined the relationship among genetic variants, DNA methylation, and gene expression in three cell types of umbilical cord samples from 204 newborn babies and found that the relationship between DNA methylation and gene expression across individuals has a different process from that across genes with in a genome. The inter-individual relationship is much less clear in terms
of negative regulation. Both active and passive roles are played by DNA methylation in regulating gene expression.

Unlike genome-wide DNA methylation, the inter-individual relationship between genetic variants and gene expression in human populations has been well-studied in both eQTL identification (*Deelen et al., 2015*) and gene expression prediction (*Xie et al., 2017*; *Zeng, Zhou & Huang, 2017*). Predicted gene expression is also used as an instrument in genome wide association studies to reduce multiple testing and identify associated genes (*Gamazon et al., 2015*). Similar studies in DNA methylation is lacking since previous studies were mostly limited to association analyses between single CpG site methylation and gene expression. It is not known whether DNA methylation of a gene has enough prediction power when all CpGs are considered together to serve as a surrogate for gene expression or enable gene expression to be an instrument in genome wide methylation studies in human populations. In this study, we examine the DNA methylation and gene expression relationship in three large human datasets. We determine the overall relationship between DNA methylation and gene expression across individuals for each gene and evaluate the predictive potential of DNA methylation data for gene expression. We also demonstrate that a penalized regression improves the overall prediction.

# METHODS

## Datasets

### Adipose dataset

This dataset is from the MuTHER study, consisting of 856 female European-descent individuals enrolled in the TwinsUK Adult Twin Registry. The quartile normalized gene expression and DNA methylation data from subcutaneous fat were downloaded from ArrayExpress (http://www.ebi.ac.uk/arrayexpress/). The gene expression data (accession number E-TABM-1140) were generated for 25,160 genes using Illumina HumanHT-12 v3.0 on 825 individuals. The log2-transformed signals were quantile normalized for each tissue followed by quantile normalized across the whole population (*Grundberg et al., 2012*). The DNA methylation data (accession number E-MTAB-1866) were generated using Illumina Infinium Human Methylation 450 from 649 female twins. The methylation beta values were already quantile normalized for each type of probe, ranging from 0 (unmethylated) to 1 (total-methylated).

### PBMC dataset

This dataset was downloaded from Gene Expression Ominbus (GSE40736). It includes 194 inner-city children with 97 cases of atopy and persistent asthma and 97 healthy controls. All the study participants were 6 to 12 years old from African American, Dominican-Hispanic and Haitian-Hispanic background (*Yang et al., 2015*). DNA methylation data were generated using Illumina's Infinium Human Methylation450k BeadChip. The normalized methylation M value matrix was downloaded from ftp://ftp.ncbi.nlm.nih.gov/geo/series/GSE40nnn/GSE40576/matrix/. Gene expression data were generated for 23,612 genes using Nimblegen Human Gene Expression arrays ($12 \times 135$ k). The normalized data matrix was downloaded from ftp://ftp.ncbi.nlm.nih.gov/geo/series/GSE40nnn/GSE40732/matrix/.

According to the publication, one outlier sample has been removed after principle component analysis, SWAN normalization was used for methylation data. Log2 transformation and RMA normalization were used for gene expression data. For each gene, expression level was standardized across samples.

### LCL dataset

The LCL dataset was generated from Lymphoblastoid cell lines (LCL) of 280 healthy individuals (96 Han Chinese-American, 96 Caucasian-American and 95 African-American). Data were downloaded from GSE23120 and GSE36369. Gene expression microarray data were generated using Affymetrix Human Genome U133 Plus 2.0 Array, which contains 38,500 well-characterized human genes covered by 54,000 probe sets (https://www.affymetrix.com/support/technical/datasheets/human_datasheet.pdf) DNA methylation data were generated using Infinium HumanMethylation450 BeadChip platform. Quantile normalized M values were used in the analyses.

## Dataset cleaning and filtering

To assess the DNA methylation effect in prediction gene expression, we defined the "methylation probes" as the 344,303 probes in Table S1 of Grundberg's (*2012*) paper. The probes on the methylation array but excluded from Table S1, which have potential SNP effects or cross hybridization effects, are termed "S&C probes". The combinations of these two types of probes are termed "all probes". The Adipose dataset has 32,478 missing values in the DNA methylation data. Samples with missing values were excluded from regression analysis of the respective gene. Among the 485,679 probes in the dataset, 344,201 probes remained in Adipose dataset after filtering. For the PBMC dataset, 344,180 out of the 485,461 probes in the dataset remained after filtering and 344,202 out of 485,578 probes remained for the LCL dataset. In order to make the method comparable and the analysis consistent, only genes that have the LASSO models were used, 8040 genes and 149,152 CpG sites in Adipose dataset, 4,252 genes and 73,553 CpG sites in PBMC dataset and 7514 genes and 143,599 CpG sites in LCL dataset (Table S1).

## Modelling the relationship between gene expression and DNA methylation

CpG probes were mapped to genes using UCSC RefGene annotation. Gene expression and DNA methylation data for each gene were extracted using in-house perl script. Since there was no missing value for methylation of the PBMC and LCL dataset, all samples were used in the regression analysis.

We used three types of regressions, single linear regression, multiple regression, and least absolute shrinkage and selection operator (LASSO) regression (*Tibshirani, 1996*), to model the linear relationship between gene expression and DNA methylation. Squared correlation ($R^2$) between predicted and observed data was used to compare the three types of regressions. In the single linear regression, each CpG site was modeled separately to predict gene expression level. The CpG site that provides maximum $R^2$ was used to represent each gene. In multiple regressions, all the CpG sites in each gene were used as predictors and the $R^2$ was calculated. In the LASSO regression with default parameters,

all CpG sites were used to predict the gene expression. We used the GLMNET package in R to fit the LASSO model in which penalized parameters were obtained using 10-fold cross-validation to minimize the mean squared error, while the predictors and response variables were all standardized.

## Cross validation

In addition to calculating the $R^2$ from fitting the models (fitting $R^2$), we also conducted five-fold cross validation to compare the prediction power of the three regression models using the validation $R^2$ ($R^2$.cv). Specifically, the samples were randomly separated into training set (4/5 of data) and testing set (1/5 of data). The procedure was iterated 10 times and the mean $R^2$ of the 10 five-fold cross validations was used as our final cross validation $R^2$ for each model. For single regression, cross validation was conducted for each CpG site and the maximum $R^2$ was used for each gene. For LASSO regression analysis, we first obtained the optimal penalty parameter using ten-fold cross validation and then used another five-fold cross validation to evaluate the predictive performance of the model.

Note that we calculated fitting $R^2$ in the LASSO cross validation models. We used the entire datasets as testing in the LASSO cross validation models in order to obtain the fitting $R^2$ in a fashion consistent with the multiple and single cross-validation models. In this case, all the $R^2$ values in the paper are squared correlation of the predicted and the true values in the training set.

## Model comparisons on significant genes

We first identified genes that showed overall model prediction $p$ values less than 0.0001 in multiple regressions and then compared the three regression models on these genes.

## Gene Ontology (GO) and pathway enrichment analysis

For top 2,000 genes with highest $R^2$, we use The Database for Annotation, Visualization and Integrated Discovery (DAVID ) at https://david.ncifcrf.gov/ (*Huang, Sherman & Lempicki, 2008*) to conduct GO term enrichment analysis based on modified Fisher Exact Test. The background genes were set to be the genes on the expression array, HumanHT-12_V3_0_R2_11283641_A. The significantly overrepresented GO terms were selected based on the EASE Score, which is the geometric mean of p-values on logarithm scale for the member terms. We applied medium classification stringency in the DAVID website to our data. "GOTERM_BP_FAT" was used to obtain more information in biological processes of the Gene Ontology enrichment analysis. "KEGG_PATHWAY" was selected for pathway enrichment analysis in the same fashion. The most enriched GO terms and pathways with low $p$-value or FDR were shown in the results.

## Gene expression prediction using different type of probes on the methylation microarray

The probes excluded by Table S1 of Grundberg's (*2012*) paper were treated as probes with SNP and/or hybridization effects (S&C probes). We compared these probes, the methylation probes, and the combination of these two types of probes in predicting gene expression.

## Analysis codes

We wrapped up our major analysis codes into a package at https://github.com/dorothyzh/MethylXcan. It includes all three regressions and calculates the squared correlation for each model. The package is written in R and Perl, and has been tested under linux or MACSOX system. Users can use this package on the datasets described here or on their own data after formatting their methylation data, expression profiling data, and annotation files as specified by the package.

## RESULTS

Association between single CpG methylation and gene expression is often conducted in human populations when both transcriptome and methylome are profiled. In this study we set out to find whether combining all CpG sites in a gene can better predict the gene expression in a human population. We obtained three human datasets, an Adipose dataset generated from subcutaneous fat tissue, a PBMC dataset from Childhood Asthma study, and a lymphoblastoid cell line (LCL) dataset. To evaluate the predicting power of DNA methylation on gene expression, we conducted three types of linear regression analyses, single regression, multiple regression, and LASSO regression for each gene. Squared correlation ($R^2$) was used for model comparisons. To focus on DNA methylation effect, we first left out CpG probes that overlap SNPs or cross-hybridize to multiple locations (S&C probes). In addition, since some genes fail to establish a LASSO model due to the lack of predictive information in DNA methylation, we only focus on genes with valid LASSO models for comparing different regression methods. In the three datasets, the total number of genes varies from 26,736 to 32,946 after quality control and normalization. About 1/6 to 1/3 of these genes have valid LASSO models with slightly bigger numbers when S&C probes are included (Table 1). In general, a large fraction of the genes with LASSO models have prediction $R^2$ greater than 0.1, but the number of genes quick reduces to hundreds and tens when $R^2$ increases to 0.2 and 0.3 (Table 1).

## Multiple regressions using all methylation CpGs from a gene predict gene expression the best in model fitting

As a reference, we first conducted association analysis on each methylation probe in predicting gene expression using single regression. Most of the genes with valid LASSO models have at least one significant CpG at nominal significance level of 0.05. For example, in the Adipose dataset, 7,326 out of 8,040 genes have at least one CpG site significant at 0.05 level and 3460 out of 8040 genes have at least one CpG site significant at 0.0001. However, the prediction power represented by the largest $R^2$ in each gene is generally low. Only 19 genes have $R^2$ greater than 0.3 and 486 genes have $R^2$ larger than 0.1 when the most predictive CpG site is considered (Table 1). Similar results were obtained from the PBMC and LCL datasets, except that the PBMC dataset has a substantially smaller number of genes with a CpG significant at 0.0001 level (582 out of 4,252 genes) although the distribution of the estimated $R^2$ is similar to that from the Adipose dataset. This could be due to the smaller sample size or the nature of the PBMC tissue source. On the other hand, the LCL

**Table 1** The number of genes with prediction $R^2$ larger than thresholds in single, multiple and LASSO regressions.

| | Dataset | Regress model | Model fitting $R^2$ | | | Cross validation $R^2$ | | | Genes w/ LASSO model | All genes |
|---|---|---|---|---|---|---|---|---|---|---|
| | | | >0.1 | >0.2 | >0.3 | >0.1 | >0.2 | >0.3 | | |
| Methylation probes | Adipose | Single | 486 | 87 | 19 | 106 | 16 | 2 | 8,040 | 26,736 |
| | | Multiple | 2,178 | 476 | 116 | 722 | 166 | 38 | | |
| | | LASSO | 1,702 | 360 | 113 | 827 | 179 | 42 | | |
| | PBMC | Single | 851 | 108 | 14 | 381 | 33 | 4 | 4,252 | 31,030 |
| | | Multiple | 3,358 | 1,163 | 382 | 746 | 109 | 30 | | |
| | | LASSO | 2,382 | 561 | 142 | 1,022 | 165 | 30 | | |
| | LCL | Single | 1,753 | 465 | 126 | 419 | 82 | 21 | 7,514 | 32,946 |
| | | Multiple | 5,138 | 2,170 | 975 | 1,663 | 575 | 185 | | |
| | | LASSO | 4,246 | 1,740 | 805 | 2,030 | 751 | 258 | | |
| All probes | Adipose | Single | 591 | 115 | 33 | 103 | 21 | 5 | 8,864 | 26,736 |
| | | Multiple | 3,037 | 760 | 211 | 898 | 226 | 64 | | |
| | | LASSO | 2,283 | 536 | 178 | 1,008 | 259 | 76 | | |
| | PBMC | Single | 1,330 | 212 | 58 | 666 | 90 | 34 | 5,064 | 31,030 |
| | | Multiple | 4,455 | 1,902 | 694 | 994 | 197 | 64 | | |
| | | LASSO | 3,207 | 870 | 235 | 1,465 | 289 | 66 | | |
| | LCL | Single | 1,888 | 533 | 155 | 425 | 88 | 32 | 7,498 | 32,946 |
| | | Multiple | 5,870 | 2,573 | 1,267 | 1,757 | 627 | 221 | | |
| | | LASSO | 4,646 | 2,029 | 999 | 2,155 | 840 | 335 | | |

**Notes.**

All genes, the total number of genes in a dataset after quality control and normalization; Genes w/LASSO model, the number of genes with valid LASSO models; All probes, the combination of methylation probes and probes with cross-hybridization/SNP effects.

dataset has a larger number of genes with higher $R^2$ from single CpG regression analysis, which could be related to the homogeneous nature of cell lines.

Since multiple CpG sites from each gene were assayed on the methylation microarray, we applied multiple linear regression to utilize all methylation CpG sites in the gene region as predictors simultaneously. The $R^2$ explained by the regression model did improve substantially for the majority of genes compared with that from the single linear regression (Fig. 1). As expected, the significant genes from multiple regression analyses tend to have larger $R^2$ compared with the non-significant genes. The improvement of $R^2$ from the multiple regression over single regress in the PBMC and LCL datasets is similar to that in the Adipose dataset (Fig. 1).

Compared with multiple regressions, LASSO regression did not generate $R^2$ quite as high in all datasets (Figs. 1D–1F), which is also indicated by smaller number of genes with $R^2$ exceeding each threshold (Table 1). For example, the number of genes with $R^2$ greater than 0.2 decreased from 476 to 360 for the Adipose dataset, from 1,163 to 561 for the PBMC datasets, and from 2,170 to 1,704 for the LCL dataset. Similar trend was observed at the other two thresholds for all three datasets.

It is widely recognized that gene expression is negatively correlated with DNA methylation level in the promoter region but often positively correlated with DNA methylation level in gene body (*Ball et al., 2009*; *Wu et al., 2010*; *Jones, 2012*). The different directions of correlation among CpG sites in the same gene may lead to perceptions that
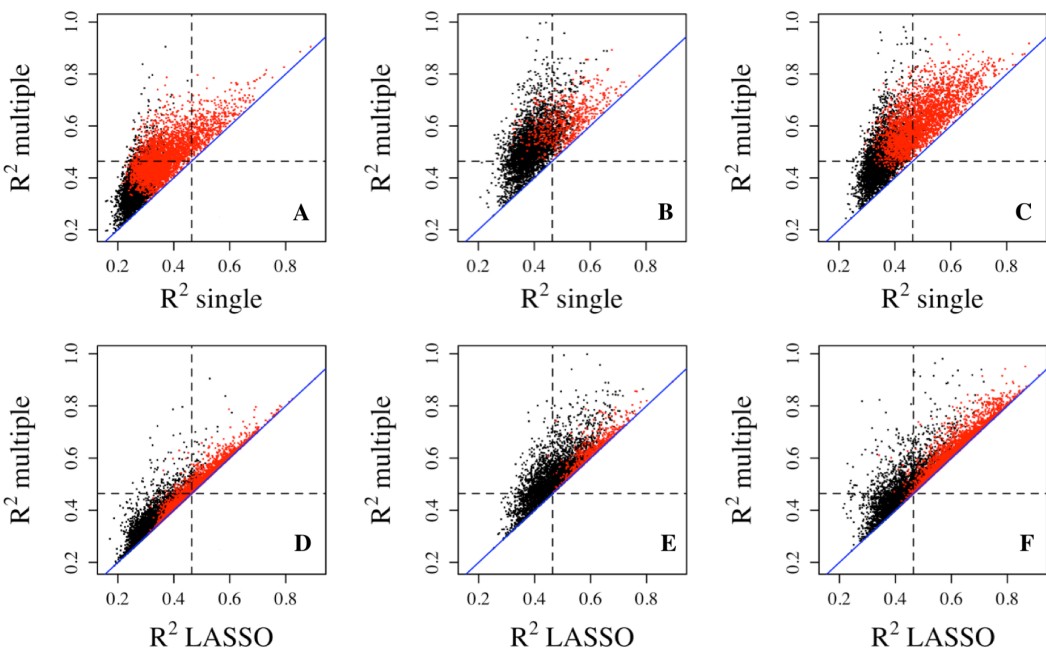

**Figure 1  Goodness of fit R² comparison among three regression models.** $R^2$ values from multiple regression are compared to those from single (top three panels) and lasso (bottom three panels) regressions in three datasets, Adipose (A, D), PBMC (B, E) and LCL (C, F). $R^2$ values shown here are on cubic root scale for visualization clarity. "single", single linear regression with the most significant CpG site as predictor; "multiple", multiple regression with all methylation CpG sites in a gene as predictors. Red points represent significant genes from multiple regressions at significance level of 0.0001. Blue solid line is the identity line and the dashed lines represent $R^2$ of 0.1.

combining all CpG sites is not advantageous in the prediction of gene expression. However, the multiple and lasso regressions can accommodate coefficients in different directions without affecting prediction power. Nevertheless, we tested CpG sites from promoter region and those from gene body for prediction separately in the LCL dataset. As expected, neither performs as well as combined (Fig. S1).

## LASSO regression shows better prediction in cross-validation

To better assess the accuracy of the predictive models, we performed 5-fold cross validation on single, multiple regressions, and LASSO regressions to estimate the prediction $R^2$. The results showed that the LASSO regression produced much larger $R^2$ values than the single regression and less dramatic but discernible increases over multiple regressions (Fig. 2). These differences are also reflected in the number of genes with $R^2$ exceeding the three thresholds. For example, 827 genes (10.29%) from the Adipose dataset have $R^2$ greater than 0.1 from LASSO regression, while 722 genes (8.98%) and 106 genes (1.32%) have $R^2$ greater than 0.1 from multiple regression and single regression, respectively (Table 1). For genes with $R^2$ greater than 0.3, LASSO regression has 42 genes (0.52%) while multiple regression and single regression have 38 (0.47%) and 2 genes (0.02%), respectively. These results indicate that penalized regression has better prediction than multiple or single regressions in cross-validation. Cross validation tends to overcome bias and over-fitting
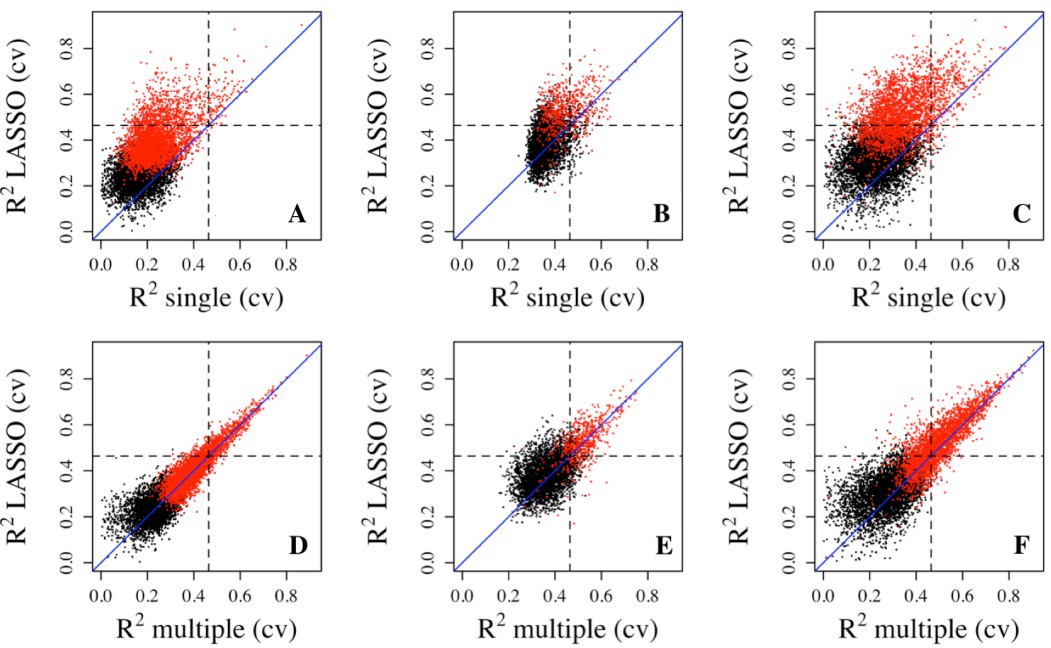

**Figure 2** **Prediction $R^2$ comparison among regression models in cross validation.** Cross-validation $R^2$ from LASSO regression are compared to those from single regression (top three panels) and multiple regression (bottom three panels) for three datasets, Adipose (A, D), PBMC (B, E) and LCL (C, F). Five-fold validation was used for all regression models. $R^2$ shown here are on cubic root scale for visualization clarity. The red points represent the significant genes from multiple regressions ($p < 0.0001$). Blue solid line is the identity line and the dashed lines represent $R^2$ of 0.1. single.cv, cross-validation $R^2$ of single regression; multiple.cv, cross-validation $R^2$ of multiple regression; cross-validation $R^2$ of LASSO regression.

issues. As expected, cross-validation $R^2$ values are generally lower than those from the model fittings, which is reflected by the smaller number of genes with $R^2$ values greater than the $R^2$ thresholds (Table 1). Similar results were obtained from the PBMC dataset and the LCL dataset.

To make sure that the prediction $R^2$ is larger than thoses from random chance, we compared the cumulative $R^2$ from the three datasets with those from the the null distribution of correlations based on Fisher z-transfromation in quantile–quantile plots (Fig. 3). All datasets showed that the observed $R^2$ values are much larger than the expected $R^2$ values from random chance. In addition, the departure is the largest in the LCL dataset followed by the Adipose dataset and the PBMC dataset when methylation probes were considered, indicating that the LASSO models capture a larger proportion of the transcriptome variability in the LCL dataset than in the other two datasets. This is potentially due to the combination of sample size and nature of different tissues.

To rule out the possibility that prediction $R^2$ is mainly driven by the variability of gene expression and the variability of DNA methylation across individuals in the study population, we first examined the correlation between the variability of gene expression with $R^2$ from LASSO regression. No obvious correlation was observed (Fig. S2). For assessing the correlation between DNA methylation variability and prediction $R^2$, we took

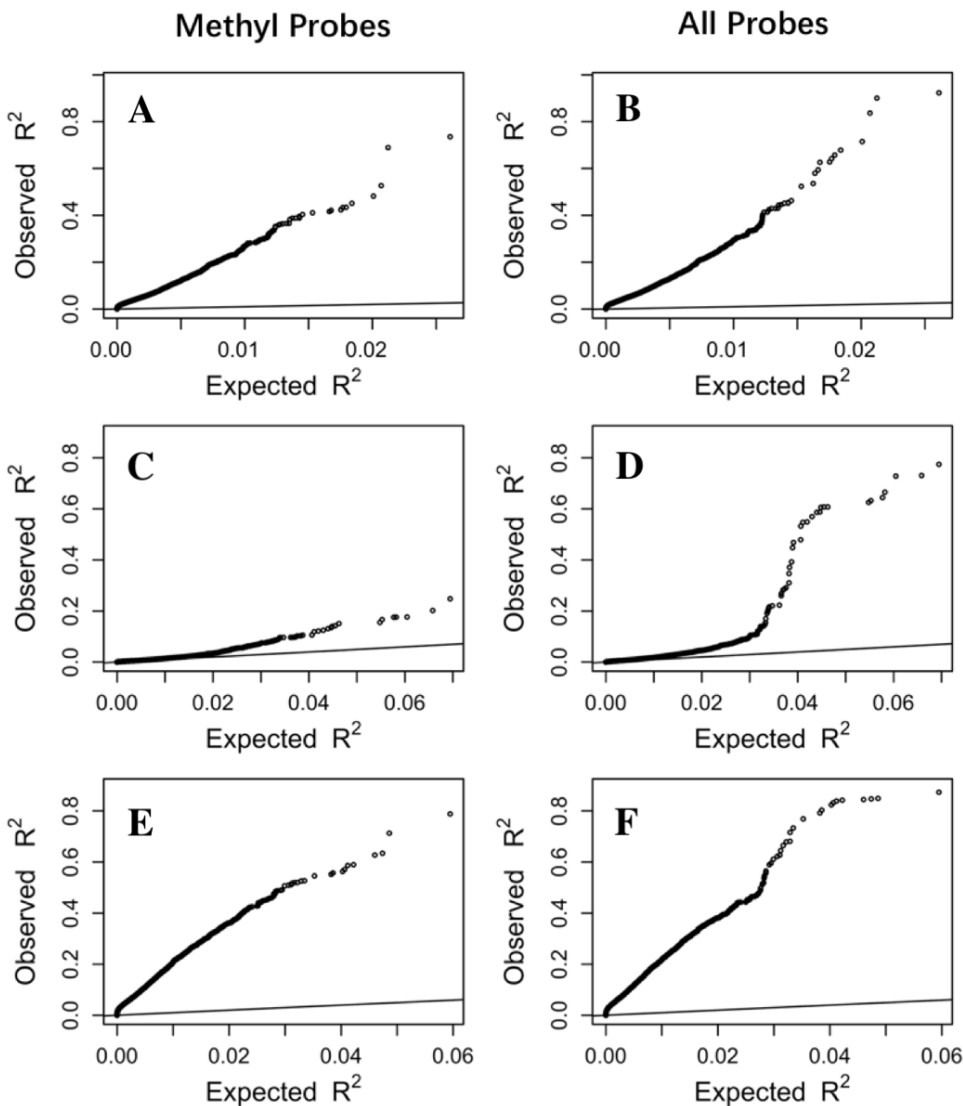

**Figure 3** **The prediction R² is beyond random chance.** Sorted $R^2$ values from three datasets, Adipose (A, B), PBMC (C, D) and LCL (E, F), are compared with those from the null distribution of $R^2$ based on Fisher z-transformation (straight line). (A, C, E) are from methylation probes, after excluding probes that have cross-hybridization or SNP effects. (B, D, F) are from all probes, the combination of methylation probes and probes with cross-hybridization/SNP effects. Five-fold cross validation was used for LASSO regression models.

the CpG with maximum $R^2$ from single regression and examined the correlation between its variability with the prediction $R^2$ from LASSO regression. We only observed a potential positive correlation in the Adipose dataset when the $R^2$ is greater than 0.5, where there are a small number of genes (Fig. S3).

## Using all probes improves prediction power for gene expression

In order to evaluate DNA methylation power in predicting gene expression, we first left out a large proportion of probes potentially affected by genetic or cross-hybridization effects

(Table S1). However, using all probes on the array is preferred if our goal is to achieve better prediction accuracy of gene expression. To evaluate the prediction power from all probes, we included all available probes in LASSO regression and found that the overall prediction power did increase compared to the models using only the methylation probes (Fig. 3). We observed more genes with $R^2$ values exceeding the thresholds (Table 1). In addition, the largest $R^2$ value is much larger when all probes are used compared to that from only the methylation probes. For example, the largest $R^2$ is 0.92 from all probes compared to 0.74 from only methylation probes in the Adipose dataset. Similarly, the largest $R^2$ increases from 0.71 to 0.88 in the PBMC dataset and from 0.76 to 0.87 in the LCL dataset. Furthermore, valid LASSO models are available for more genes when all probes are used (Fig. S4).

The increase of prediction power on gene expression from all types of probes on the methylation microarray suggests that there is contribution from the probes with potential SNP effects or cross hybridization effects. To further assess the size and nature of their contribution, we separately estimated the prediction power of the methylation probes, S&C probes, and the combination of them (all probes). The results showed that the S&C probes have independent prediction power from the methylation probes and the combination of both has increased prediction power over the methylation probes alone (red points vs black line in Figs. 4A, 4C and 4E). The prediction power from the S&C probes was also estimated for genes with enough SNP probes to form a LASSO model and their prediction power are mostly above zero (blue points in Fig. 4). The fact that the blue points are randomly distributed instead of following the black line suggests that the two sources of $R^2$ are not correlated; therefore, the genetic effect and epigenetic effect do not seem to coexist in the same genes. Figure 5 shows some examples of genes with large prediction powers from either methylation probes or S&C probes. As expected, the methylation probes tend to show continuous methylation values while the S&C probes tend to show categorical values due to limited genotypes of the samples.

## GO term analysis of better predicted genes

To examine the potential biological function of the genes showing relatively higher predictability, we conducted gene ontology (GO) enrichment analysis using DAVID on genes with $R^2$ larger than 0.2 from LASSO regression of methylation CpGs. At false discovery rate (FDR) of 0.01, cell adhesion, lipids metabolism, and regulation of immune system are among the most significantly enriched terms in the Adipose dataset (Table S2), which seem to be consistent with the previous findings for subcutaneous fat cells (*Berg & Scherer, 2005*). For the PBMC dataset, the most significantly enriched terms are mostly related to defense and immune functions, lymphocyte aggregation, T cell activation, inflammatory response, as well as cell adhesion. These results appear to be reasonable for atopy and persistent asthma blood cells. For the LCL data, some terms related to cell adhesion, migration, communication, and morphogenesis are highly enriched. Same GO term analyses were also conducted for $R^2$ from all CpG probes and similar results were obtained.
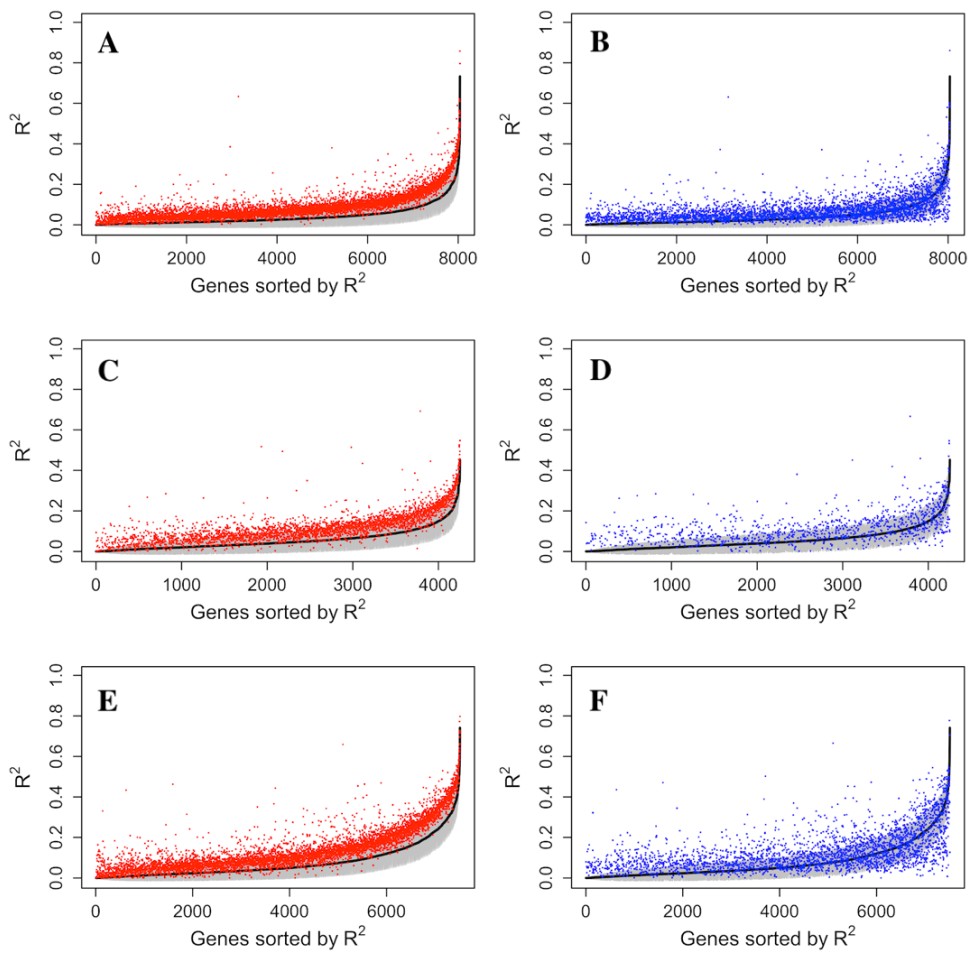

**Figure 4   Comparison of R² from using methylation probes, S&C probes and all probes.** LASSO regression $R^2$ values from three datasets, Adipose (A, B), PBMC (C, D) and LCL (E, F), were generated from methylation probes (black line), S&C probes (probes with cross-hybridization/SNP effects) (blue points), and all probes (the combination of methylation probes and probes with cross-hybridization/SNP effects) (red points). The 95% confidence interval of $R^2$ from methylation probes is shown as a grey shadow.

## DISCUSSION

We examined the relationship between gene expression and DNA methylation across the genome using data from three large human studies. We explored three linear regression models for predicting gene expression and found that shrinkage based LASSO multiple regression provides the best prediction. However, even with LASSO regression, the methylation probes can predict expression in only a small proportion of genes with moderate prediction power. We also demonstrated that using all probes on the methylation array does improve prediction power to some degree.

Three types of regression models were examined in our study for their prediction power evaluated by squared correlation ($R^2$). The single linear regression is based only on the best predictive CpG in each gene, therefore, has least prediction power. The

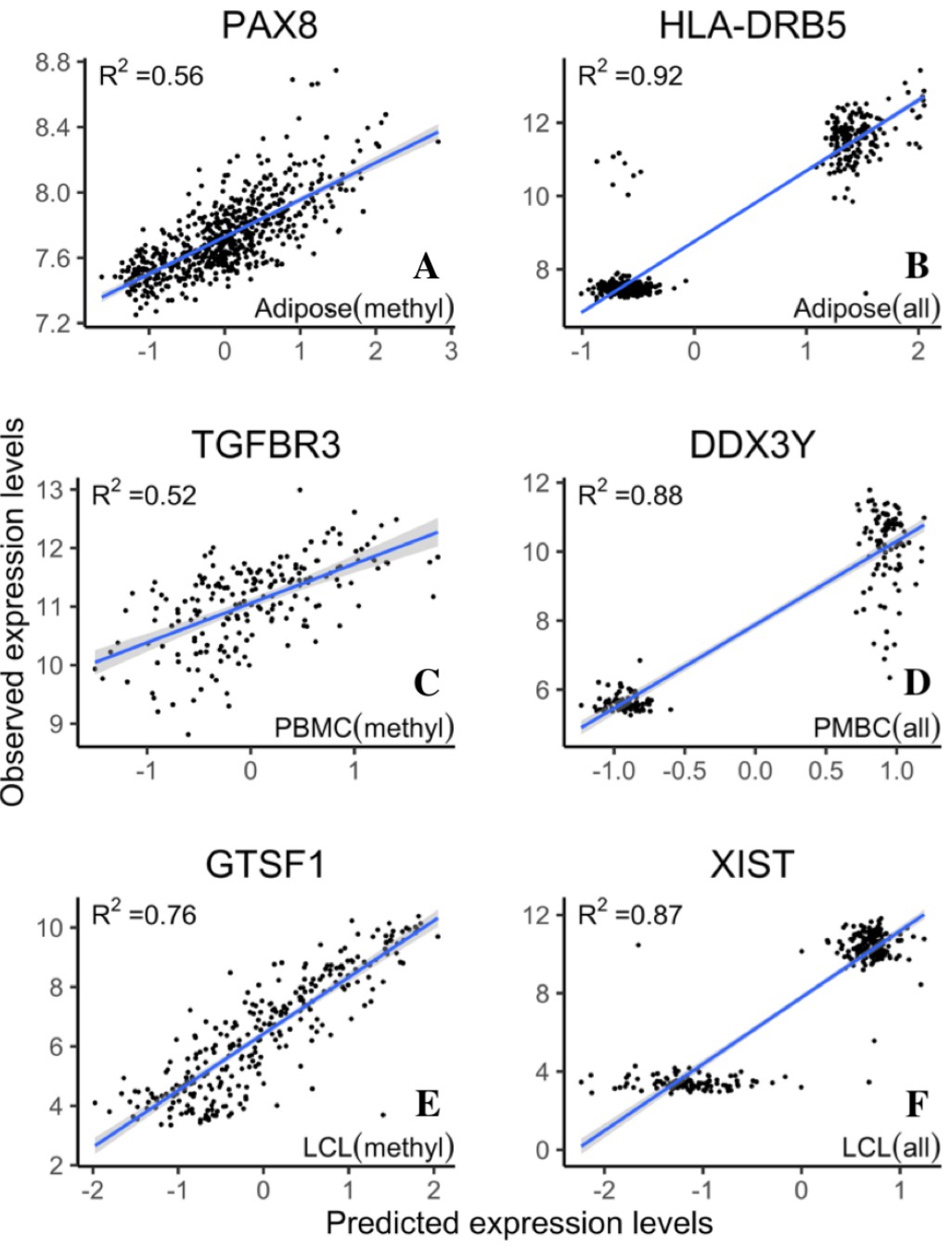

**Figure 5** **Example genes with high prediction power.** $R^2$ is from LASSO regression models. Adipose, PBMC and LCL are the three datasets. $X$-axis indicates the predicted expression levels from LASSO regression models; $y$-axis indicates observed expression levels for each dataset (methyl, methylation probes; all, all probes).

multiple regression has increased power when all CpG sites in each gene are included as predictors; however, it has substantial over-fitting problem for genes with large number of CpGs. The shrinkage based LASSO regression overcomes the over-fitting problem

without losing predictability. LASSO imposes sparsity among the coefficients and puts constraint on the overall absolute values of the regression coefficients, which forces certain coefficients to be zero. This property is beneficial in avoiding model overfit as well as variable selection and model interpretability. In this study, not all expressed genes have LASSO models because LASSO fails to select informative predictors in some genes even with minimum penalization, which indicates that no predictive information exists in the DNA methylation data at these genes. LASSO is not the only shrinkage-based regression method. There are other penalty regression models, such as the Ridge (*Hoerl & Kennard, 1970*), elastic net (*Zou & Hastie, 2005*), elastic net with rescaled-coefficients and grouped lasso (*Yuan & Lin, 2006*; *Meier, Van De Geer & Bühlmann, 2008*). Further evaluation is needed for their merits in improving prediction in this setting.

The prediction power from DNA methylation in our analysis seems to be much lower than that from DNA sequence variants evaluated in different human tissues (*Gamazon et al., 2015*). One potential reason for relative low prediction power we observed from DNA methylation is the complex mechanisms of gene expression regulation. In addition to DNA methylation, transcription factors, histone modification (*Verdin & Ott, 2015*), and non-coding RNAs (*Janowski et al., 2005*; *Ting et al., 2005*; *Ting, McGarvey & Baylin, 2006*; *Kaikkonen, Lam & Glass, 2011*) all play critical roles in gene transcription regulation (*Jones, 2015*). Some more comprehensive tools, such as FEM (*Jiao, Widschwendter & Teschendorff, 2014*) and ROADMAP (*Kundaje et al., 2015*), may help integrate the influences of the other factors on gene expression. Another potential reason for low prediction power of methylation is that the landscape of DNA methylation differs dramatically across cell types, tissues (*Lokk et al., 2014*), ages (*Teschendorff et al., 2010*), and races (*Song et al., 2015*). The relationship between gene expression and DNA methylation could also vary substantially across these factors. The correlation between gene expression and DNA methylation from bulk studies at population level encompasses all these variabilities; therefore, it is not surprising to see lower prediction power in the PBMC and adipose datasets compared to the LCL dataset. The potential of DNA methylation alone as surrogate for gene expression is likely to be limited in general, especially in the tissues with mixed cell types, such as PBMC, which is used widely in human population studies. The combination of DNA methylation and genotype, should be more powerful for this purpose, as indicated by the increased prediction power when SNP-containing probes were included in the prediction models (Figs. 3 and 4). This can be a promising future direction.

## CONCLUSIONS

We explored three regression methods to predict gene expression using DNA methylation, single regressions, multiple regressions, and LASSO penalized regression. LASSO regression reduces over-fitting and improved the prediction power. All three datasets we analysed show relatively low prediction power. The better predictive genes are dataset specific and their function varies in different tissues or cell types. Overall, we will recommend caution for using one's methylation profile to predict one's transcriptome.

## List of Abbreviations

| | |
|---|---|
| **SNP** | single nucleotide polymorphisms |
| **LASSO** | least absolute shrinkage and selection operator |
| **R2** | squared correlation |
| **S&C** | the probes have potential SNP effects or cross hybridization effects |
| **MuTHER** | Multiple Tissue Human Expression Resource Project |
| **PBMC** | peripheral blood mononuclear cell |
| **LCL** | lymphoblastoid cell |
| **MDS** | myelodysplastic syndrome |
| **GO** | gene ontology |

### Funding

Degui Zhi was partially supported by NIH Grant R01 HG008115; Xiangqin Cui was partially supported by NIH 2P60AR048095. Huan Zhong was supported by Hong Kong Baptist University's strategic development fund SDF15-1012-P04 to Yiji Xia. There was no additional external funding received for this study. The funders had no role in study design, data collection and analysis, decision to publish, or preparation of the manuscript.

### Grant Disclosures

The following grant information was disclosed by the authors:
NIH: R01 HG008115, 2P60AR048095.
Hong Kong Baptist University's strategic development: SDF15-1012-P04.

### Competing Interests

Degui Zhi and Xiangqin Cui are Academic Editors for PeerJ. The authors declare there are no competing interests.

### Author Contributions

- Huan Zhong analyzed the data, prepared figures and/or tables, authored or reviewed drafts of the paper, approved the final draft.
- Soyeon Kim analyzed the data, authored or reviewed drafts of the paper, approved the final draft.
- Degui Zhi and Xiangqin Cui conceived and designed the experiments, authored or reviewed drafts of the paper, approved the final draft.

### Data Availability

For the Adipose Dataset, the gene expression data is available at E-TABM-1140; and the DNA methylation data can be found at E-MTAB-1866. For the PBMC Dataset, gene expression data is available at GSE40732; and the DNA methylation data is available at GSE40576. For the LCL Dataset, gene expression data is available at GSE23120; and the DNA methylation data is available at GSE36369. The software package 'MethylXcan' is available at https://github.com/dorothyzh/MethylXcan.

## Supplemental Information

Supplemental information for this article can be found online at http://dx.doi.org/10.7717/peerj.6757#supplemental-information.

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
