# Peer review of "Predicting gene expression using DNA methylation in three human populations"

_PeerJ, doi:10.7717/peerj.6757_

## Round 0.1 · original submission · Major Revisions

The reviewers found that the research problem was interesting and well defined. The analyses were deemed to be methodologically and statistically sound. However, both reviewers raised concerns about the conclusion drawn based on only two datasets. More datasets may be needed to validate the method and support the conclusion. The differences from existing tools should also be discussed. Please revise or rebut according to the reviewers’ comments.

Reviewer 1 ·

Basic reporting

The article describes a method of imputing gene expression values from DNA methylation data by using two publically available data sets. However, the prediction R2 for most genes is quite low, which makes this method potentially less useful for eQTL or differential expression studies.

However, the paper fits well into the scope of peerj as the quality of the article by describing a methodologically and statistically sound analysis, regardless of subjective determinations of 'impact,' 'novelty' or 'interest'.

Experimental design

General Points

1. Selection of publically available data: There are many more publically available data sets (e.g. from HapMap LCLs). It would be useful to test the method in more tissues, as well as perhaps having a replication cohort in the same tissue. It’s interesting that one is a very heterogeneous tissue with multiple cells, and one data set is of a purified single cell.

2. There should be at least mention of some previous attempts to impute gene expression data from genotypes, as well as predicting methylation sites from nearby methylation sites.

3. Could the accuracy be further improved by classifying methylation sites by location relative to the gene, position relative to CpG islands, mean methylation, as well as variability in methylation? Especially in the multiple regression model.

Validity of the findings

4. The author mentions that “the combination of DNA methylation and other predictors, such as genotype, may be more powerful for this purpose”. Both data sets have genotypes, it would be possible to test this hypothesis (maybe this is outside the scope of the paper)

5. It’s interesting that immune genes come up as being best predicted - is this driven by the PMBC results? One explanation here could also be that there is a bias towards genes relevant to immune cells as there is more variability in these genes (as opposed to other genes completely switched off)

Additional comments

Specific Comments:

Line 51: Could mention CpG islands, shores. While 70% of CpGs are methylated, many of the CpGs are clustered into CpG islands, which are mostly unmethylated. The regions around the CpG islands have been found to have high variability in methylation.

Line 69: “They found that the association between gene expression and methylation is not always negative in promoter region or positive in gene body.” Need to mention where this assumption comes from, why was it expected that the gene body methylation would be positively correlated with expression? Could mention this study that look into the causal relationship of DNA methylation with expression https://www.ncbi.nlm.nih.gov/pmc/articles/PMC3673336/

Line 73: “asthma”

Line 86: Do you mean “quantile normalized” instead of “quartile normalized”?

Line 117: How many samples remain after removing those with missing values? This means that only samples with 100% call rate were kept?

Line 125: Is this the RefGene annotation by Illumina? What are the cutoff for considering a probe as belonging to a gene, is there a cis-interval cutoff?

Line 138: what are the parameters used in glmnet?

Line 142: Have you considered using a completely separate “testing” data set to avoid overfitting?

Line 157: Multiple Testing correction?

Line 159: I think the GO enrichment studies need to be carried out using a matched permutation test instead of the whole array background (for example like it is implemented in eForge). This is because some of the probes might already be biased to come up in any kind of analysis in the specific cell types, as they have more variation, or a certain context to genes or CpG islands

Line 224: Perhaps mention that previously you just considered the R2 in model fitting, without worrying about overfitting to your data sets, so the multiple regression performs best here, whereas the LASSO model is best when considering prediction.

Line 282: “the larger genetic effect and larger epigenetic effect” - both are larger than what?

Line 283. This is interesting, it seems that the S&C “predictive power” is based around the separation of two distinct clusters. Surprising that there aren’t three cluster based on the three genotypes AA, AB and BB for each SNP - does this happen to for the S&C probes? I could imagine this also happening for some of the probes in “all” as they might have missed some in the Grundberg analysis. Is there a way to quantify how often this clustering phenomenon happens? (e.g. through running a clustering algorithm over all probes). Right now it looks like just four examples were cherry-picked.

Line 287: Again, this might not be surprising as the genes associated with GO terms relevant to fat and blood are more likely to be expressed, and thus provide more variability?

Line 314: I think a better method might be to compare the R2 in an independent validation set?

Reviewer 2 ·

Basic reporting

The study by Zhong et al is trying to find correlation between gene expression using DNA methylation.

Experimental design

Please provide link to perlscript used in the study.

Validity of the findings

Conclusions are too strong and based on only two studies.

Additional comments

The study by Zhong et al address role of DNA methylation in predicting gene expression. The manuscript introduction is clear and Figures are relevant. The authors have done commendable job in defining the research question. The major problem is for such a strong conclusion the authors did not provide any validation of their analysis and have only analyzed two study. I think it is very important to combine role of Histone modifications and enhancers/repressors and other gene regulatory mechanisms in addition to DNA methylation for gene expression analysis.
1. The authors use DNA methylation of all CpG sites to predict gene expression. This is obvious since not CpG sites will lie in promoter region. It is known that CpG-dense promoters of actively transcribed genes are never methylated, but reciprocally transcriptionally silent genes do not necessarily carry a methylated promoter.
1. The authors should discuss how MethylXcan is unique from existing softwares, such as FEM, ROADMAP Epigenomics and others.
2. The authors indicate DNA methylation is poor predictor of gene expression but develop MethylXcan program to predict gene xpression using DNA methylation.
3. One of the major problem of their results is lack of validation. Result from cancer genome atlas project indicate correlation between DNA methylation and gene expression (Long et al 2017). To remove any bias the authors should analyze atleast 5-6 other datasets and include the results from those analysis.
4. Although the results described by Zhong et al are very interesting and can have profound impact on epigenetics. The authors should compare correlation between gene expression and DNA methylation at either Promoter region or on total gene body.
5. The authors indicate immune and defense gene show correlation between DNA methylation and gene expression. However there is no information provided about role of Histone modifications, DNA binding elements, transcription factors. The author should discuss how methylation of immune and defense gene is unique and related details in detail.
6. Figure 5 : The axis are not labelled
7. Line 126 Please provide link to perlscript

8. The English language usage and formatting issues in the manuscript.
a. Line 93 “Humanmethylation450”
b. Line180 “Thereinto”
c. Line210 “less than identifies”
d.Table 2 legend “andKEGG”

e. Figure 5 legend “Examplegenes”

f. Figure 4 legend “Comparisonof”

g. Figure 2 legend “Comparisonamong”

h. Line 100 reference formatting error “background [9]”
i. Line 50 “residue(You”
j. Line 59 “2013)studied”

---

## Round 0.2 · accepted · Accept

The reviewers are satisfied with the revision. Thanks for the hard work to make such substantial improvement.

# Reviewer 2 ·

Basic reporting

The article structure, figures, tables is professional. Authors indicate that Raw data for perlcode is shared in GITHUB.

Experimental design

Research question is well defined.

Validity of the findings

Data is robust, statistically sound, & controlled.

Additional comments

I am overall satisfied with author response and revised manuscript. I do not have any further comments.